# Neural Architecture Search by Learning a Hierarchical Search Space

## Abstract

Monte-Carlo Tree Search (MCTS) is a powerful tool for many non-differentiable search related problems such as adversarial games. However, the performance of such approach highly depends on the order of the nodes that are considered at each branching of the tree. If the first branches are not discriminative enough, i.e. they cannot distinguish between promising and deceiving configurations for the final task, the efficiency of the search is exponentially reduced. While in some cases the order of the branching is given as part of the problem (e.g. in chess the sequential order of the moves is defined by the game), in others, such as Neural Architecture Search (NAS), the visiting order of the tree is not important, and only the final architecture matters. In this paper, we study the application of MCTS to NAS for the task of image classification. We analyze several sampling methods and branching alternatives for MCTS and propose to learn the branching by hierarchical clustering of architectures based on their similarity. The similarity is measured by the pairwise distance of output vectors of architectures. Extensive experiments on two challenging benchmarks on CIFAR10 and ImageNet show that MCTS, if provided with a good branching hierarchy, can yield promising solutions more efficiently than other approaches for NAS problems.

## 1 Introduction

Neural Architecture Search (NAS) aims to automate neural architecture design and has shown great success in the past few years (Zoph & Le, 2016; Real et al., 2019; Liu et al., 2018a; Ren et al., 2021), outperforming manually designed Convolutional Neural Networks (CNN) in deep learning (Liu et al., 2018b; 2019; Guo et al., 2020). NAS aims at yielding the best architecture from a given search space with a lower computational budget than a brute-force approach, based on training all possible architectures independently.

One prominent solution is one-shot methods based on weight sharing (Pham et al., 2018), in which multiple architectures share all or part of their weights, eliminating the need for training architectures individually to evaluate their performance. In this approach, a "supernet" that contains all operations/architectures is trained and each architecture is evaluated using weights inherited from that supernet. When architectures are compatible, this approach allows to recycle training iterations (Cha et al., 2022; Pham et al., 2018; Bender et al., 2018); however, when architectures are vastly different, it could lead to interference, i.e the weights that are good for one architecture are detrimental to another and vice-versa (Roshtkhari et al., 2023).

To reduce the effect of interference in weight sharing, previous work has used either multiple models that can focus on different parts of the search space (Roshtkhari et al., 2023; Su et al., 2021b; Zhao et al., 2021a; Hu et al., 2022); or importance sampling (Liu et al., 2018b; Ye et al., 2022; Xu et al., 2019) in which the probability of an architecture performing well is estimated during training. In the latter, promising architectures are sampled more frequently during the course of training, gradually reducing the possible interference among architectures (Liu et al., 2018b; You et al., 2020) as the training is guided towards the best architectures. Unlike methods that use uniform sampling for training the supernet (Guo et al., 2020; Roshtkhari et al., 2023), the challenge of using importance sampling is to robustly estimate and identify superior architectures as soon as possible in the training cycle, to enable the model to focus on them and to avoid wasting training resources on unpromising

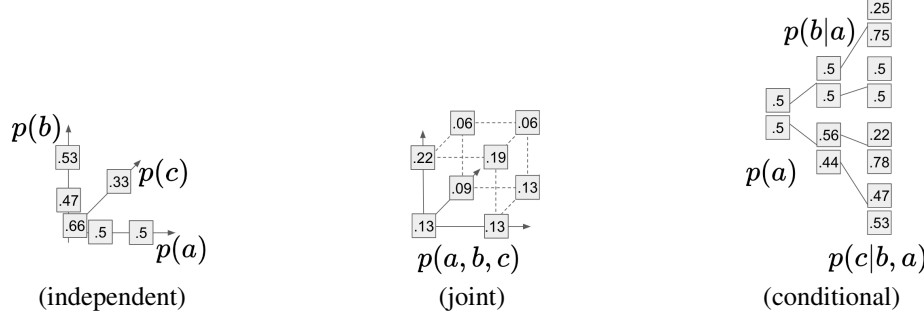

Figure 1: **Probability factorization of 8 architectures.** We show different ways to approximate the discrete probability distribution of architectures for a toy example of search space with N=3 nodes (a,b,c in the figure) each one with O=2 possible operations for a total of $2^3$ architectures. (left) Assuming the nodes independent (as in DARTS (Liu et al., 2018b)) allows the model to estimate only $N \times O$ probabilities. (center) Considering the joint probabilities would require to estimate $O^N$ different probabilities (as in Boltzmann sampling). (right) The joint probability can be factorized into the product of conditional probabilities (in a hierarchy such as in MCTS). This does not reduce the probabilities to estimate, but allows a more efficient exploration of the search space.

architectures. This requires fast and reliable estimation of the probability distribution of architectures in as few training iterations as possible.

A neural network can be considered as a graph, composed of nodes, which define the architecture of the network, connected by edges. These nodes have a choice of operations, which are the actual processes applied to the data (e.g. convolution, fully connected, etc.). In importance sampling, an assumption that makes architecture probability estimation more efficient is "node independence", i.e. considering nodes as statistically independent variables. For instance, in a neural network, the choice of the operation for the second layer would not depend on the choice of the operation in the first layer. As the result, the estimation of an architecture is approximated as the product of nodes' probabilities (see Fig. 1 (independent)). This reduces the scale of the problem to learning individual node probabilities. While the widely used differentiable NAS method (DARTS (Liu et al., 2018b) and followup works (Xu et al., 2019; Li et al., 2020a; Ye et al., 2022)) use this assumption, overlooking the joint contribution of nodes to architecture performance can lead to a poor node selection for the final architecture (Ma et al., 2023).

In order to remove the node independence assumption, the joint probabilities of all configurations should be estimated (see Fig. 1 (joint)). This requires estimating the probability of each architecture independently. Considering mainstream single-path one-shot (SPOS) methods (Guo et al., 2020; Chu et al., 2021b; Li & Talwalkar, 2020; Stamoulis et al., 2019), at each iteration, one architecture from the supernet is sampled, trained and estimated and that estimation is used to update the probability distribution. With node independence, at each iteration several associated node probabilities will be updated, while for joint probabilities only one probability associated with the sampled architecture would be updated. Thus, the full update of the estimation cost proportional to the number of architectures, making it unscalable to large search spaces. Consequently, inefficient estimation of promising architectures slows down the entire training as the importance sampling will not be able to focus on the right architectures.

To explore more wisely, a compelling option is to factorize the joint probability into conditionals, such as a tree used in Monte-Carlo Tree Search (MCTS) (see Fig. 1 (conditional)). While the number of probabilities to estimate do not decrease compared to joint probabilities, this factorization can bring some important advantages. A good hierarchical organization of search space allows more efficient traversal of the tree by reducing the unnecessary exploration of unpromising branches. Prioritizing branches with good solutions can lead to faster convergence, better solution quality and improved scalability. To maximize these advantages, the early nodes of the tree should be highly discriminative. Standard predefined hierarchies of architectures do not guarantee a more efficient exploration, as they are defined by construction and without taking into account the semantic similarity of the corresponding architectures.

While in many MCTS problems the searching hierarchy is defined by the sequentiality of the problem (e.g the moves in chess), for NAS there is no constraint in the order in which the architectures are explored. Zhao et al. (2021b) and Wang et al. (2021a) leverage this by using the classification accuracy of the nodes for partitioning the search space into "good" and "bad" nodes to reduce the unnecessary exploration. This approach works well when running the search with a fixed, already learned, recognition model (i.e the CNN weights). In fact, Zhao et al. (2021b) uses MCTS for searching the best performing model on a supernet pre-trained with uniform sampling, while Wang et al. (2021a) perform MCTS for NAS using the already trained models provided by NAS-Bench-201 (Dong & Yang, 2020).

In this work, we tackle the more challenging problem of learning the recognition model and the tree partitioning jointly. In this setting, at the beginning of the optimization, the recognition model would have a poor performance, and using its accuracy as proxy to partition the search space would not work well. We propose instead to estimate the distance between architectures with a partially trained recognition model in a unsupervised way, without considering the model accuracy. The output vector of each architecture, sampled from this supernet, is used to calculate a pair-wise distance matrix of architectures. We propose to use this matrix for hierarchical clustering and generating the tree partitions. The resulting hierarchical clustering implicitly enforces that the early nodes of tree to be semantically related, without directly considering their performance. This allows for a better learning and consequently makes the search faster.

The main contributions of our work are the following:

- We present a new understanding of classical choices of models and approaches for NAS based on the sampling approach and the estimation of the underlying probability of a given architecture. We show that too restrictive assumptions (e.g. node independence) allow for faster training, but converge to worse solutions. Instead, more realistic assumptions could lead to better solutions if used with some additional regularization.

- We propose to learn to sample from a MCTS efficiently by learning a good hierarchy that avoids low performing architectures. We evaluate several approaches to build this hierarchy and show that the most promising one is obtained through a measure of pairwise distances among architectures derived from a supernet pre-trained with uniform sampling.

- We empirically validate our findings on two NAS benchmarks on CIFAR10 dataset and mobilenet ImageNet search space, showing that the proposed approach is very general and can find promising architectures within a limited computational budget.

## 2 RELATED WORK

**One-shot methods.** One-shot methods (Pham et al., 2018; Bender et al., 2018) has become very popular in NAS (Liu et al., 2018b; Guo et al., 2020; Su et al., 2021a) due to their efficiency and flexibility. Generally, the training of supernet and searching for best architecture can be decoupled (Guo et al., 2020; Wang et al., 2021a) or performed simultaneously (Liu et al., 2018b). In the former, the search can be performed by various methods, such as random search (Bender et al., 2018; Li & Talwalkar, 2020), evolutionary algorithms (Guo et al., 2020) or MCTS (Wang et al., 2021a) and the supernet is static during this phase. The latter alternates between training the supernet and updating the reward to guide the search, such as updating architecture weights in differentiable methods (Liu et al., 2018b), controller in RL (Pham et al., 2018) or probability distribution in MCTS (Su et al., 2021a). The quality of supernet as a proxy for architecture evaluation has been the subject of scrutiny in recent years, with various results in different settings (Yu et al., 2019; Wang et al., 2021b; Zela et al., 2019; Termritthikun et al., 2021). A proposed solution is to explicitly reduce the weight sharing among architectures by non-hierarchical factorization of the search space (Zhao et al., 2021a; Roshtkhari et al., 2023; Su et al., 2021b). However, in general these methods are computationally more expensive as they require training additional models. Tree based approaches can be viewed as a form of hierarchical factorization of the search space that reduces weight sharing.

**Node independence.** While some earlier NAS methods using reinforcement learning (Zoph & Le, 2016; Pham et al., 2018), or evolution (Real et al., 2019; Sun et al., 2020; 2019) do not treat nodes independently, they were computationally expensive. More efficient and widely used NAS methods are differentiable methods (based on DARTS (Liu et al., 2018b)) that use back-propagation to learn

node (architecture) weights (probabilities) alongside supernet weights. However, one of their known problems is that the learned weights of the nodes do not accurately reflect their contribution to the ground truth performance and ranking (Wang et al., 2021b; Yu et al., 2019). While several works has recognized and tried to improve DARTS (Chen et al., 2021c; Ye et al., 2022; Xu et al., 2019), few have directly explored the contribution of node independence assumption to this problem (Ma et al., 2023; Xiao et al., 2022).

Shapley-NAS (Xiao et al., 2022) highlights the underlying relationship between nodes by showing that the joint contribution of node pairs differs from the accumulation of their separate contribution, due to their possible collaboration/competition. They propose reweighing the learned architecture weights by utilizing the Shapley value. However, the estimation of the Shapely value can be costly as it requires training supernet multiple times. ITNAS (Ma et al., 2023) proposes to explicitly model the relationship between the nodes by introducing a transition matrix and an attention vector that denotes the node probability translation to successor nodes. The matrix and vector are optimized in a bi-level framework alongside node probabilities. However, the application of this method is limited to only cell level (inner layer operations) and the application to a more general macro search space is not straightforward (for details about macro and cell-based search spaces see Appendix A). While these works try to incorporate node dependencies into differentiable NAS, an alternative approach is to attempt to directly learn either joint or conditional probabilities of the sampled architectures.

**Monte-Carlo Tree Search.** MCTS with Upper Confidence bound applied to Trees (UCT) (Auer et al., 2002) has been used previously for NAS (Negrinho & Gordon, 2017; Wistuba, 2017). AlphaX (Wang et al., 2019) used a surrogate network to predict the performance of sample architectures, and MCTS to guide the search. TNAS (Qian et al., 2022) aims to improve the exploration of the search space by using a bi-level tree search that traverses layers and operations iteratively. However, the binary tree that factorizes the operations is manually designed.

LaMOO (Zhao et al., 2021b) and LaNAS (Wang et al., 2021a) aim to tackle the problem of finding the best architecture and assume that deep leaning model is given either from a trained supernet (Wang et al., 2021a) or using precomputed benchmarks (Zhao et al., 2021b). In our work instead we aim at training the deep learning model and finding the corresponding optimal architecture with MCTS in the same optimization, which makes the problem more challenging.

The closest work that jointly performs the model training and architecture search with MCTS is Su et al. (2021a) that propose to construct a tree branched along operations. During the training, a hierarchical sampling is used for node selection, updating the supernet weights and the reward (training loss). Node statistics are then used to update a relaxed UCT probability distribution. However, the tree design is manual and a regularization method is required to compensate for insufficient visits of nodes.

## 3 TRAINING BY SAMPLING ARCHITECTURES

Our training is based on a SPOS (Guo et al., 2020) in which, given a neural model $f$ (e.g. a CNN) for each mini-batch of training data $\mathcal{X}$ and corresponding annotations $\mathcal{Y}$, a different architecture $a$ from the search space $\mathcal{S}$ is sampled and back-propagated with the following loss:

$$\mathcal{L}(f_a(\mathcal{X}, w), \mathcal{Y}) = \sum_{(x,y) \in (\mathcal{X}, \mathcal{Y})} l(f_a(x, w), y), \tag{1}$$

where $l$ is the loss for a sample (for instance cross-entropy) and $w$ is the network weights. The training speed and model performance can drastically vary depending on how $a$ is sampled. For importance sampling methods, to avoid overfitting on training data, we use validation accuracy as the reward to estimate the probability distribution; and use on-line estimation on mini-batches to accelerate the process. In the following sub-sections, we present some of the most common sampling techniques, from uniform sampling to our proposed approach.

**Uniform sampling.** This is the simplest and the original approach of SPOS (Guo et al., 2020), in which the architecture is sampled uniformly: $a \sim \mathcal{U}(|\mathcal{S}|)$, where $|\mathcal{S}|$ represents the cardinality of the search space. Although very simple, this sampling is unbiased (does not privilege specific architectures), so that, given enough training, all architectures will have the same importance. This method does not require to store any information during the training, and in principle it can work with

any $|\mathcal{S}|$, even very large ones. In practice however, the equal importance of architectures can lead to two possible problems that hinder the quality of the solution: i) With strong weight sharing (i.e. most of the model weights are shared among all configurations), the same weight would have to adapt to very different architectures,leading to destructive interference and therefore low performance (see Roshtkhari et al. (2023)). ii) If architectures do not share many weights, each is almost independent and the training time would increase proportionally to $|\mathcal{S}|$. While uniform sampling can be combined with search space partitioning to find a trade-off (Roshtkhari et al., 2023), it demands training multiple models, therefore higher memory consumption and computational cost. A different direction is to find ways to prioritize the sampling of the more promising architectures.

**Importance sampling with independent probabilities.** The simplest way to estimate the importance of each operation is by assuming each node $a_i$ as independent. Thus, the probability of an architecture $a$ is approximated as $p(a) = p(a_1)p(a_2)...p(a_t)$. This simplifying assumption allows for a better sampling efficiency by factorizing the number of probabilities to estimate. However, the quality of the solution is decreased, as it disregards the joint influence of nodes on the performance (Ma et al., 2023).

**Importance sampling with joint probabilities: Boltzmann sampling.** In Boltzmann sampling, architecture $a$ is sampled from a Boltzmann distribution with probability $p(a) \propto \exp(\frac{\epsilon_a}{T})$, where $\epsilon_a$ is estimated rewards (here accuracy) of $a$, and $T$ is the temperature. Sampling is performed with an annealing temperature, starting from a high temperature (almost uniform), such that the initial phase of the training is unbiased, to a low temperature (almost categorical) such that the training focuses on high performing architectures. While more efficient than uniform sampling, estimating $\epsilon_a$ is still time consuming, especially for large search spaces and it is difficult to balance exploration/exploitation trade-off in Boltzmann exploration (Cesa-Bianchi et al., 2017).

**Sampling with conditional probabilities: Tree Search.** Instead of considering a flat vector of probabilities, we consider a tree of conditional probabilities: $p(a) = p(a_t|a_{(t' \leq t-1)})p(a_{t-1}|a_{(t' \leq t-2)})...p(a_1)$. Each $a_t$ represents a level of the tree and separates the set of possible architectures into disjoint subsets. The commonly used structure of the tree (see Fig.2 (b) Default Tree Structure) is defined by factorizing the model architectures layer by layer (Su et al., 2021a), starting from the first to the last one. Assuming for simplicity a symmetric binary tree, the first level would split the configurations into two disjoint groups, which would be recursively separated into two groups at each following level. With uniform sampling, at each iteration the nodes in level $t$ are sampled with probability $(1/2)^t$. Thus, the estimations of the probabilities of early nodes would be sufficiently accurate because of high sampling rate. In contrast, for Boltzmann, the sampling probability is $1/|\mathcal{S}|$, which can be very small for a large search space. However, if the first nodes maintain a near uniform probability distribution (not discriminative enough), the estimation of the posterior nodes would suffer from low sampling rate. A possible solution is the regularization proposed by Su et al. (2021a), in which at each update of a node, other equivalent nodes (at same level with same operation) are updated similarly with an exponential moving average. This allows for multiple updates simultaneously, that allows for more rapid estimation of probabilities and more efficient exploration.

While this solution seems to work adequately well, this regularization comes with limitations: i) It assumes that the tree is homogeneous at each level, i.e. each node has a similar structure (same children) as others at the same level, which limits the approach to only certain kind of search spaces. For instance, this approach would not work for search spaces in which the operations in a node are conditioned to the choice of operation at the previous node. ii) The assumption of reusing the same probabilities for equivalent nodes, implies treating nodes independently. In this case the node independence assumption is enforced in a soft way by a regularization coefficient. Thus, the method tries to find a compromise between full independence and conditional dependence, but it is unclear if this trade-off is optimal.

## 4 OUR APPROACH

In our approach, we also tackle the low sampling rate issue in posterior nodes of a MCTS, but from another perspective. We aim to find an ordering of nodes for the tree such that, especially for the first levels of the tree, the probabilities of a sub-node are imbalanced. In this way, the model is able to focus on a reduced set of architectures as the unpromising branches would be estimated early and

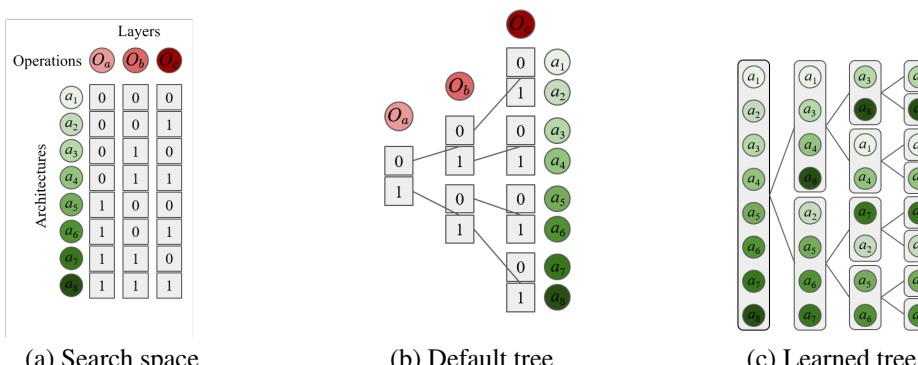

(a) Search space          (b) Default tree          (c) Learned tree

Figure 2: **Comparison of the standard tree structure and our learned structure on a 3 binary operations search space.** (a) The search space consists of architectures with 3 binary operations $(o_a, o_b, o_c)$ which leads to 8 architectures $(a_1, a_2, ..., a_8)$. (b) The default tree structure uses the order of operations (e.g. layers) to build the tree, however this is not optimal. (c) Our learned tree structure uses a tree that is generated by an agglomerative clustering on the model outputs.

sampled less frequently. Consider for instance the example in Fig. 1 (conditional): if instead of building the tree from node $a$, we would start from node $c$, as the conditional probabilities of $c$ are imbalanced, the model would be able to focus on a good branch more effectively, thus making the sampling much more efficient. This is possible because, in contrast to other problems in which the ordering of nodes is defined as part of the problem, in NAS the final architecture matters and not the order used to reach it. Therefore, in this work we propose to learn an improved ordering of nodes to sample in a MCTS, instead of using a predefined ordering.

**Tree design.** Consider the toy search space shown in Fig.2(a), with 3 binary operations $(O_a, O_b, O_c)$ which leads to 8 different architectures. The default tree design (Fig. 2(b)) as presented in sec.3, branches off the tree on operation choices per layer. In this section, we present a different approach to build a tree of architectures. As shown in Fig.2 (c), our tree is built as a hierarchical clustering of architectures. Essentially, each node of the tree is a cluster of architectures, going from the root that contains all architectures in a single cluster to the leaves that each contains a single architecture. Through this approach, we release the dependency of the tree from the binary operations and allow any possible hierarchical grouping of configurations. In this work, we build a hierarchy that keeps into account the semantic similarity of among different configurations A representation of architectures that is independent of the quality of supernet, while adequately summarizes the relevant information for the task, can be used to determine the placement of architectures in search space. A clustering algorithm can then be used to build the hierarchy based on the difference in distances between architectures. We note that while the ultimate goal of NAS is to find the architecture with highest accuracy, the supernet accuracy is an inadequate representation of architecture for this purpose. The output vector on the other hand, is more suitable as distances between architectures would have semantic meaning in the class space (See sec.5.2 for ablation studies on tree design).

First, a supernet is pre-trained with uniform sampling for a number of iterations. We then sample architectures from the supernet and perform a forward pass with one mini-batch of validation data and record the output vector. The pairwise distances of architectures are calculated and the resulting distance matrix and is used for hierarchical agglomerative clustering (Murtagh & Legendre, 2014) to construct a binary tree. We argue that this method allows us to effectively cluster architectures that have similar overall functionality, even if they might differ in their structure in term of their operations. For more details about the construction of the tree, see algorithm 1 in Appendix B.

**Search and training.** We use a variation of MCTS for both supernet training and architecture search. Similar to Su et al. (2021a); Wang et al. (2021a), in our case, the tree is already fully expanded and thus expansion and roll-out stages of traditional MCTS are skipped. Similar to Su et al. (2021a), we use Boltzmann sampling for the selection stage. The Boltzmann distribution allows sampling proportional to the probability of the reward function, producing better exploration (Painter et al.,

2024), which is fundamental for a good training of the model. For a node in the tree $a_i$, we perform importance sampling with a Boltzmann sampling relative to the node:

$$p(a_i) = \frac{\exp(R(a_i)/T)}{\sum_j \exp(R(a_j)/T)}, \tag{2}$$

where $R$ is our reward function and $T$ is temperature, determining the sharpness of distribution and the normalization sum on $j$ is on the sibling nodes of $i$. Training consists of sampling each level of the tree from the root to the leaf, followed by a gradient update of the recognition model $w$ with the sampled architecture $a$ on a mini-batch of training data $\mathcal{X}_{tr}$ and an update of the reward function for the explored nodes, from the leaf to the root to the tree based on the obtained accuracy of the architecture on a mini-batch of the validation set $\mathcal{X}_{val}$. To balance exploration/exploitation, we use the Upper Confidence bound applied to Trees (UCT) (Kocsis & Szepesvári, 2006) as reward for sampling. Considering a node in tree $a_i$, our reward is defined as:

$$R(a_i) = C(a_i) + \lambda\sqrt{log(|parent(a_i)|)/|a_i|}, \tag{3}$$

in which the second term is to explore the search space. We use function $|a_i|$ to show number of times node $a_i$ is visited, with the constant $\lambda$ that controls the exploration/exploitation trade-off. Here, node $parent(a_i)$ indicates the parent node of $a_i$. We define $C$ as:

$$C(a_i) = \beta\,C(a_i) + (1 - \beta)\,Acc(f_a(\mathcal{X}_{val}, w)), \tag{4}$$

which is a smoothed version of the validation accuracy for the used architecture $a$ , with smoothing factor $\beta$, to account for the noisy on-line estimation on mini-batches. For further details about the training algorithm see algorithm 1 in Appendix B. To search for final architecture after training, we sample $k$ architectures without exploration ($\lambda = 0$) and rank them based on their performance on validation dataset, selecting the best one as final architecture.

## 5 EXPERIMENTS

In this section, we perform experiments on CIFAR10 dataset (Krizhevsky et al., 2009) using two macro search space benchmarks ,and ImageNet (Russakovsky et al., 2015) with MobielNetV2-like (Sandler et al., 2018) search space . We compare our proposed method with several various sampling methods discussed in sec. 3 (see the summary in Table 6 in Appendix B). In all experiments, we use SPOS method for sampling and training the supernet. For MCTS methods, we start by uniform sampling of the architectures, and after a warm-up period, we use the recorded statistics to calculate UCT (equation 3) and sample using equation 2. Moreover, we initialize $C(a_i) = 1$ to further ensure more exploration at the early stages of training as $C(a_i) < 1$ when nodes are visited. During training, the average accuracy of the supernet will improve, balancing the exploitation/exploration. For MCTS default tree, used by Su et al. (2021a), each layer of CNN is considered as a level of tree, with operations providing the branching and compare the method with and without soft node independence assumption (regularization). For more details on the experiments see Appendix B.

### 5.1 POOLING SEARCH SPACE

To fully investigate our proposed method, we use Pooling search space, which is a small (36 architectures) but challenging CIFAR10 benchmark with Resnet20 (He et al., 2016) architecture. The only architecture parameter that is searched is feature map sizes at each layer (or equivalently which layer to perform downsamplings). Therefore, at each layer the choices are whether to perform downsampling (pooling operation) or not (Identity operation). The main challenge of this search space is the full weight sharing of architectures that contributes to the inadequacy of several common search methods(Roshtkhari et al., 2023).  We represent architectures with number of layers per feature map sizes (e.g. [4,3,3] meaning 4 layers in high resolution, 3 layers middle resolution and 3 layers in low resolution). Our method achieves better results compared to others with similar or less search time (Table 1). For MCTS (default tree design), the regularization proposed in Su et al. (2021a) seems to help, however our method obtains better performance without needing regularization.

Table 1: **Accuracy and ranking on the Pooling benchmark on CIFAR10.** We report the found architecture (represented with number of layers per feature map sizes), best and average of 3 training accuracy and ranking and search time for different methods.

| Method | Arch. | Best Acc. | Avg. Acc. | Best Rank | Avg. Rank | Search Time |
|---|---|---|---|---|---|---|
| Default Arch. | [4,3,3] | 90.52 | - | 15 | - | - |
| Uniform | [4,3,3] | 90.52 | $90.40 \pm 0.08$ | 15 | 17 | 1.5 |
| MCTS | [4,4,2] | 90.85 | $90.57 \pm 0.21$ | 12 | 15.3 | 2 |
| Boltzmann | [3,5,2] | 90.88 | $90.51 \pm 0.12$ | 11 | 15.3 | 3 |
| Independent | [3,5,2] | 90.88 | $90.86 \pm 0.01$ | 11 | 11.7 | 2 |
| Mixtures (Roshtkhari et al., 2023) | [5,3,2] | 91.55 | $91.36 \pm 0.27$ | 4 | 5 | 6 |
| MCTS + Reg.(Su et al., 2021a) | [6,1,3] | 91.78 | $91.42 \pm 0.11$ | 3 | 3.6 | 2 |
| MCTS + Learned (ours) | [6,2,2] | 91.83 | $91.72 \pm 0.12$ | 2 | 3 | 2.2 |
| Best | [7,1,2] | 92.01 | - | 1 | - | - |

Table 2: **Distance measures for the similarity matrix.** We compare the final performance of our MCTS with learned tree structure which is built with an agglomerative clustering using a similarity matrix between network outputs with different distance measures.

| Distance Measure | Best Arch. | Best Acc. | Average Accuracy |
|---|---|---|---|
| cross-entropy | [5,3,2] | 91.55 | $91.20 \pm 0.23$ |
| L2 | [6,2,2] | 91.83 | $91.52 \pm 0.36$ |
| KL | [6,2,2] | 91.83 | $91.72 \pm 0.12$ |

## 5.2 ABLATION STUDY

We perform several ablations to study the convergence and performance of our method and investigate alternative ways to design the hierarchy in Pooling search space.

**Tree partitioning with accuracy.** We investigate the importance of using the output vector and clustering to build the tree. For doing that, we build a tree using the accuracy obtained from a pre-trained supernet, and to recursively partition search space into "good" (top 50% of the partition) and "bad" regions (bottom 50%). Performing MCTS on this tree, we obtained best and average accuracy of 90.85% and 90.01% respectively, showing diminished results compared to our method.

**Clustering distance measures.** We use output of CNN for a mini-batch of data to calculate pairwise distances. The difference between two distributions can be calculated by several distance measures, here we investigate L2 distance, KL divergence, and cross-entropy (Table 2).

**Supernet training and similarity. matrix** During supernet training, the average accuracy of architectures increases. In our experiments we used supernet at convergence. However, since the criteria for our proposed clustering is similarity and not the exact accuracy, we only need to train supernet long enough to capture that similarity. We explore our method with various iterations of supernet training (see Fig.3(left)). We observe that even without any training, we can still find a better architecture than the default, and with only 1/3 of full training, we can find better architecture compared to Roshtkhari et al. (2023). This suggests that we can balance the supernet pertaining budget and search budget as the trade-off.

**Branching quality and NAS convergence.** To show that good branching can speed up NAS, we compare NAS with our learned tree with default tree and a binary tree created from a random matrix (see Fig. 3(right)). While the average accuracy of supernet increases in general over epochs, the branching quality can affect how the search space is explored. After a warm-up period for UCT, our tree consistently outperforms default and random tree. This suggests that the quality of branching is important in learning how to explore the search space more efficiently and a low quality hierarchy can lead to poor performance.

**Alternative branching.** While using the output matrix to design the branching is well-performing, it requires some initial training of the supernet. We explore using two types of encodings as a zero-cost proxy to calculate the similarity matrix. Representing an architecture as a graph, the general encoding

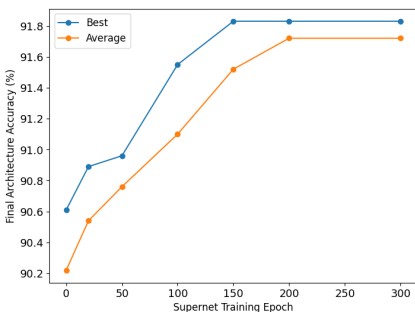 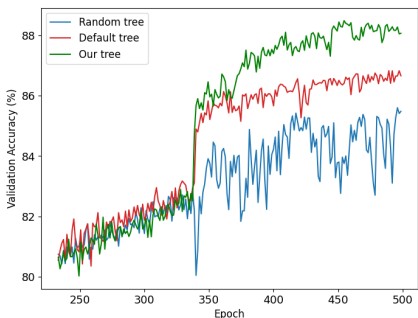

Figure 3: (left) **Training epochs for estimating the similarity matrix.** We show the final performance of our MCTS in which the tree structure is learned with a model uniformly trained for a given number of epochs. For best results at least 200 epochs are needed; (right) **Accuracy over epochs for several training strategies.** After the warm-up phase, our approach is constantly better than default tree or MCTS with a randomly selected tree.

Table 3: **Comparing various zero cost branching methods.** We consider one-hot encoding of operations per layer or the categorical vector representation. The similarity matrix is calculated using L2 distance. We also consider an exponential weighting scheme to increase the influence of earlier layers on distance.

| Encoding | Weighting | Best Arch. | Best Acc. | Avg. Acc. |
|---|---|---|---|---|
| Vector | | [2,5,3] | 90.89 | $90.63 \pm 0.68$ |
| Vector | ✓ | [3,4,3] | 90.92 | $90.81 \pm 0.15$ |
| One-hot | | [5,2,3] | 90.96 | $90.60 \pm 0.22$ |
| One-hot | ✓ | [5,1,4] | 91.05 | $90.78 \pm 0.21$ |

is the adjacency matrix, corresponding to the edges (or one-hot encoding of operations per layer). We also consider the categorical representation of the one-hot encoding as vectors as another choice. As shown in Table 3, these two zero-cost encoding techniques do not perform as well as our proposed approach. However, their performance is still better than the most common techniques shown in Table 1. We note that naively calculating the distances (same weight for all layers) is equivalent to considering each layer as an independent variable, while in fact the posterior layers have less importance than early layers. Therefore, we weight each layer $l$ exponentially with $1/2^l$, when calculating the similarity matrix, which leads to slight increases in performance. We hope that in future work it will be possible to learn a good tree structure without pre-training the supernet.

### 5.3 NAS-Bench-Macro search space

This benchmark based on MobileNetV2 (Sandler et al., 2018) blocks consist of 8 layers and operation set {*Identity*, *MB3_K3*, *MB6_K5*} resulting in $3^8 = 6,561$ (3,969 unique) architectures. Table 4 presents results of several sampling based NAS approaches. From the table, we see that our method yields the best architecture of the search space. For each method, we use the best reward and present an ablation on different rewards in Appendix C.1.

### 5.4 Search on ImageNet

ImageNet (Russakovsky et al., 2015) consists of 1.28 million training images in 1000 categories. In our experiments, we use 50k images of validation set as the test data to compare with other methods. To accelerate our training we use mixed precision and FFCV (Leclerc et al., 2023) library. We use similar macro search space to Su et al. (2021a); You et al. (2020); Chu et al. (2021b); Guo et al. (2020), based on MobileNetV2 (Sandler et al., 2018) blocks with optional Squeeze-Excitation (SE) (Hu et al., 2018) module. The total operation choice per layer is 13 resulting in $13^{21}$ search space size for 21 layers. The choices are convolution kernel size of $\{3, 5, 7\}$ and expansion ratio of $\{3, 6\}$, identity and SE option.

Table 4: **Accuracy and ranking on NAS-Bench-Macro.** We compare our method and several approaches in terms of best, average accuracy and ranking. The architectures are represented with operation index per layer.

| Sampling | Arch. | Best Acc. | Avg. Acc. | Best Rank | Avg. Rank |
|---|---|---|---|---|---|
| Boltzmann | [12220111] | 92.39 | $92.30 \pm 0.10$ | 406 | 453 |
| Independent | [22120211] | 92.44 | $92.29 \pm 0.21$ | 347 | 412 |
| MCTS | [22221210] | 92.74 | $92.51 \pm 0.18$ | 80 | 246 |
| Uniform | [21222220] | 92.79 | $92.58 \pm 0.20$ | 56 | 197 |
| MCTS + Reg. (Su et al., 2021a) | [12222222] | 92.92 | $92.67 \pm 0.18$ | 21 | 112 |
| MCTS + Learned (ours) | [22212220] | 93.13 | $92.97 \pm 0.12$ | 1 | 6 |
| Best | [22212220] | 93.13 | - | 1 | - |

Table 5: **Comparison of accuracy and computational cost on ImageNet classification task.** The architecture are searched on MobilenetV2-based search space. We consider light weight models with target budget of around 280 MFLOPs. In the top part of the table we report results of other NAS methods, while at the bottom we report results of our baselines and our approach.

| Method | Top-1 | Top-5 | FLOPs(M) | Params(M) | GPU days |
|---|---|---|---|---|---|
| MobileNetV2 1.0 (Sandler et al., 2018) | 72.0 | 91.0 | 300 | 3.4 | - |
| MnasNet-A1 (Tan et al., 2019) | 75.2 | 92.5 | 312 | 3.9 | 288 |
| SCARLET-C (Chu et al., 2021a) | 75.6 | 92.6 | 280 | 6.0 | 10 |
| GreedyNAS-C (You et al., 2020) | 76.2 | 92.5 | 284 | 4.7 | 7 |
| MTC_NAS-C (Su et al., 2021a) | 76.3 | 92.6 | 280 | 4.9 | 12 |
| Uniform | 72.2 | 89.7 | 277 | 4.6 | $\sim 5$ |
| Boltzmann | 73.1 | 89.9 | 278 | 4.7 | $\sim 5$ |
| MCTS + Reg. (Su et al., 2021a) | 76.0 | 92.6 | 280 | 4.9 | $> 12$ |
| MTCS + Learned (Ours) | 76.3 | 92.6 | 280 | 4.9 | $\sim 7$ |

Similar to Su et al. (2021a), we define a FLOPs budget for our search. We leverage the fact that FLOPs can be used as a zero-cost proxy for architecture performance (Chen et al., 2021b) and search only within a certain range of target budget ($[0.99, 1] \times$ budget) by sampling architectures and discard those not within the budget. To compare directly with Su et al. (2021a), we set a budget of 280 MFLOPs. In Table 5.4, we compare our method with several NAS approaches (taken from Su et al. (2021a)) on the upper part of the table, while we compare with our sampling based approaches on the bottom part. Note that MCTS + Reg. is our re-implementation of Su et al. (2021a), with some minor performance differences. Our method yields an architecture that provides high accuracy with a limited GPU cost. We attribute this advantage to the learned structure of the tree, that allows a quicker learning of the promising architectures.

## 6 CONCLUSION

In this work, we introduce a novel method to design a hierarchical search space for NAS. we highlight the shortcomings of node independence assumption used in popular NAS methods and the impact of hierarchical search space design on search quality and efficiency. We show that by simply learning the proper hierarchy, we can achieve state of the art results with MCTS without requiring further regularization and established our method empirically the by extensive evaluation on CIFAR10 and ImageNet.

**Limitations** Our proposed method in its current form works best in relatively small ($< 10,000$) search spaces due to the quadratic complexity of the construction of the pairwise distances for clustering. Nevertheless, by focusing on a selected budget we were able to tackle much larger search space as on ImageNet. An analysis of the complexity of our algorithm is provided in Appendix C.5.

**Reproducibility Statement** We provide general experimental details in sec. 5 and to further facilitate the reproducibility of our experiments, we provide necessary implementation detail and hyprameters in Appendix B.

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

## A  MORE RELATED WORK

**Search space design**  Common NAS search spaces can be categorized as Micro (cell-based), Macro, and mulit-level. Micro search spaces (Zoph et al., 2018; Liu et al., 2018b) focus on yielding the optimal architecture by finding the operations that can produce the best model inside a cell or block. The cell is then stacked to form the entire network, while the outer skeleton of the network is often controlled manually by including reduction cells. This was inspired by observing that the state-of-the-art manual architectures were formed by repetition of a certain structure and helped to reduce the complexity of search space to a manageable level (White et al., 2023).

Therefore, the objective of this approach is to find a cell that works well on all parts of the network, which might be suboptimal. This neglects to explore non-homogeneous architectures and diminishes the capability of NAS to find novel architectures. Macro search space (Su et al., 2021a) instead searches for the outer skeleton of the network while fixing the operations at micro level. This can include architecture parameters such as: type of layers, number of layers, or channels in layers, pooling positions etc. Finally, a mulit-level search space searches at two levels: cell and macro structure for a CNN (Liu et al., 2019) or convolution and self-attention layers for vision transformers (Chen et al., 2021a). In this work, we focus on macro search spaces as it is generally more expressive and challenging than micro search spaces.

**NAS benchmarks**  NAS benchmarks are used to evaluate for a given method the quality and the amount of computation required to yield a solution. They have played a crucial role in the NAS community as they provide an evaluation of all architectures in a brute-force way to find the optimal solution and eliminate the need to run independently this expensive process. The development of NAS benchmarks has also improved the reproducibility and efficiency of NAS research. Tabular benchmarks (Ying et al., 2019; Su et al., 2021a) are constructed by exhaustively training and evaluating (metrics such as accuracy, FLOPS, number of parameters, etc.) all possible architectures. On the other hand, surrogate benchmarks (Siems et al., 2020) estimate the architecture performance using a model which is trained on data from several trained architectures. Benchmarks have been developed for both micro (Ying et al., 2019; Dong & Yang, 2020; Siems et al., 2020) (with addition of channels (Dong et al., 2021)) and macro (Su et al., 2021a; Roshtkhari et al., 2023) search spaces. For more details on NAS benchmarks see the survey Chitty-Venkata et al. (2023).

**Architecture encoding**  Some works have shown that architecture encoding can affect the performance of NAS (White et al., 2020; Ying et al., 2019) and good encoding of architectures enables efficient calculation of relationships or distances among architectures. The most common encoding represents the architecture as a directed acyclic graph (DAG) and adjacency matrix along with a list of operations (Ying et al., 2019; Zoph & Le, 2016). For using performance predictors, BANANAS (White et al., 2021) proposes a path-based encoding instead of adjacency matrix and GATES (Ning et al., 2020) proposed a graph based encoding scheme that better mode the flow information in the network. Encoding can also be learned by unsupervised training prior to NAS often utilizing an autoencoder (Li et al., 2020b; Lukasik et al., 2021; 2022; Yan et al., 2020; Zhang et al., 2019) or a transformer (Yan et al., 2021). In our work, we make use and compare different ways of encoding architectures for our approach as ablation and show that measuring distances based on the network output seems to be the fundamental for good results.

## B  EXPERIMENTAL SETUP AND DETAILS

**Sampling method details**  A summary of various methods used in our experiments (Tables 1 and 4) is presented in Table 6.

**Training Algorithm**  The training pipeline for our method is shown in algorithm 1. First, a pre-training with random sampling of the architectures is performed in order to train an initial model $f$ with parameters $w_p$. With this model we build a pairwise matrix $D_{i,j}$ that measures the distance of configuration $i$ and $j$ on the output space of the model. With this matrix, we use agglomerative clustering to build a binary tree that represents the hierarchy that will be used for the subsequent MCTS training. During training, an architecture is sampled from the tree, where at each node Boltzmann sampling with the learned probabilities is used. Then, this architecture is used to update

Table 6: **Summary of sampling methods used in our experiments.**

| Method | Search Space Structure | Sampling Method |
|---|---|---|
| Uniform | Flat | Uniform (sec. 3) |
| Independent | Flat | Nodes sampled independently (sec. 3) |
| Boltzmann | Flat | Joint prob. (sec. 3) |
| Mixture | Flat (partitioned) | Uniform (Roshtkhari et al., 2023) |
| MCTS | Hierarchical (def. tree) | Conditional prob. ( sec. 3, Tree Search) |
| MCTS + Reg. | Hierarchical (def. tree) | Conditional prob. + regularization (Su et al., 2021a) |
| MCTS + Learned | Hierarchical (learned tree) | Conditional prob. (sec. 4) |

the model on a mini-batch of training data (for simplicity we did not include momentum in the gradient updates) and to estimate its accuracy on a validation mini-batch. The validation accuracy is smoothed with an exponential moving average and used as reward with UCT regularization for updating the node probabilities of the sampled architecture. Finally, after the MCTS, the best architectures are sampled from the tree by sampling the tree with $\lambda = 0$.

---

**Algorithm 1:** Simplified pseudo-code of our training pipeline.

---

**Input** : $\mathcal{S}$: Search Space; $\mathcal{X}_t, \mathcal{X}_v$: mini-batches of training and validation data; $f_a$: model with architecture $a$; $w_p, w$: weights of the pre-trained and final model initialized randomly; $e_{pt}, e_{MCTS}$: pre-training and MCTS iterations; $\alpha$: learning rate; $\beta$: smoothing factor; $\lambda$ : exploration parameter of UCT.

$\#pre\text{-}training$
**while** $epochs \leq e_{pt}$ **do**
    $a \leftarrow$ sample from $\mathcal{U}(|\mathcal{S}|)$
    $w_p \leftarrow \mathrm{w}_p - \alpha \nabla_{w_p} \mathcal{L}(f_a(\mathcal{X}_t, w_p))$
**end**
$\#build\ the\ search\ tree$
**for** $a^i \in \mathcal{S}$ **do**
    Output vector $o^i \leftarrow f_{a^i}(\mathcal{X}_v, w_p)$
**end**
Distance matrix $D_{ij} = dist(o^i, o^j)$
Binary tree $\mathcal{T} \leftarrow aggl\_clustering(D)$
$\#main\ training\ with\ MCTS$
**while** $epochs \leq e_{MCTS}$ **do**
    $\#sample\ an\ architecture\ a$
    $a = []$
    $node = $ "root"
    **push**($a,node$)
    **while** $not(is\_leaf(node))$ **do**
        $node \leftarrow sample(next(node))$ with Boltzmann sampling as in Eq.(2)
        **push**($a,node$)
    **end**
    $\#update\ model\ w\ and\ accuracy$
    $w \leftarrow w - \alpha \nabla_w \mathcal{L}(f_a(\mathcal{X}_t, w))$
    $accuracy \leftarrow \mathrm{Acc}(f_a(\mathcal{X}_{val}, w))$
    $node \leftarrow$ **pop**(a)
    $\#update\ rewards$
    **while** $not(is\_root(node))$ **do**
        $parent \leftarrow$ **pop(**a**)**
        $C(node) \leftarrow \beta\, C(node) + (1 - \beta)\, accuracy$
        $R(node) \leftarrow C(node) + \lambda \sqrt{log(|parent|)/|node|}$
        $node \leftarrow parent$
    **end**
**end**
**Output :** Best architecture from $\mathcal{T}$ by sampling with $\lambda = 0$

---

### B.1 Dataset and hyperparametes

For experiments performed on CIFAR10 (Kocsis & Szepesvári, 2006) dataset, we split the training set 50/50 for NAS training and validation. To tune hyperparameters, we either performed grid search or when comparing with other works used similar hyperparameters. We used SGD with weight decay and cosine annealing learning rate schedule. Furthermore, for MCTS methods we split training iterations to roughly 40/25/35 fractions for uniform sampling/MCTS warm-up/MCTS sampling respectively. In all experiments we use $\beta = 0.95$ and $\lambda = 0.5$.

**Pooling search space**  This search space introduced by Roshtkhari et al. (2023) is based on Resnet20 (He et al., 2016) architecture. The only CNN parameters to search is where to perform pooling. To calculate distance matrix we trained the supernet for 300 epoch using uniform sampling with batch size 512, learning rate 0.1 and weight decay 1e-3. For search we trained for 400 epochs with batch size 256, learning rate learning rate 0.05 and temperature $T$ is set to linearly annealing schedule (0.02, 0.0025). Since this search space is small we only consider nodes with max probabilities and report the it as the final architecture.

**NAS-Bench-Macro search space**  This benchmark introduced by Su et al. (2021a) is based on MobileNetV2 (Sandler et al., 2018) blocks. The supernet pre-training is performed for 80 epochs with batch size 512 and learning rate 0.05. For search we use batch size of 256 for 120 epochs with and $T$ linearly annealing from 0.01. At the ends of training, we sample 50 architectures from the tree and report the best as final architecture.

**ImageNet**  To accelerate our training in ImageNet experiments, we use mixed precision and FFCV (Leclerc et al., 2023) library. We sample architectures within a FLOPs budget and discard those outside of it. We train for 100 epochs with SGD and cosine annealing learning rate. Other training strategies are similar to experiments on CIFAR10.

## C  Additional results

### C.1  Reward for MCTS

The most common rewards used for NAS algorithms is accuracy and loss. While loss is differentiable, accuracy is more aligned with the objective of NAS. Furthermore, either the training or validation can be used to calculate the reward. Instead of absolute values, a relative training loss metric was used in Su et al. (2021a) to account for unfair reward comparison at different iteration of supernet training. In Table 7 for NAS-Bench-Macro, we explore some common combination of options to estimate the reward. In all setting our approach performs on par or slightly better than Default Tree + Regularization. For both methods it seems that using the accuracy on the validation set as metric is the best. However, while for our approach the best performing configuration is obtained with the absolute metric, for Su et al. (2021a) the relative metric seems slightly better.

### C.2  Distance matrices

We visualize the distance matrices calculated using output vector and various encoding in Fig. 4.

### C.3  Tree visualizations

For pooling search space, we visualize tree structure based on our proposed method and architecture encodings in Fig. 5. The tree is presented with architecture indices on leaves. The architecture indices and the corresponding performance can be found in Roshtkhari et al. (2023).

### C.4  Architecture visualization

We show the found architecture by our method for ImageNet in Fig. 6.

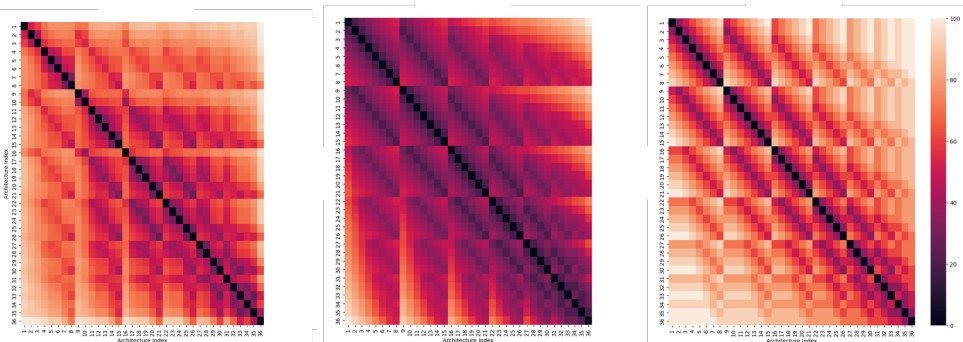

Figure 4: **Normalized distance matrices calculated with various methods.** (left) Distance matrix calculated from output vectors (our method) ; (middle) From vector encoding ; (right) From one-hot encoding. The architecture indices on leaves correspond to indices used in Pooling benchmark (Roshtkhari et al., 2023).

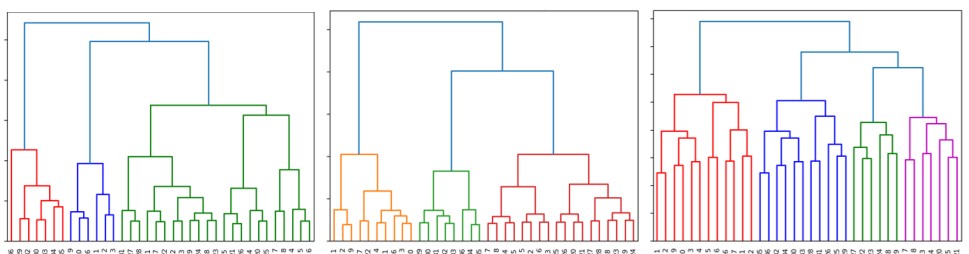

Figure 5: **Tree branching for Pooling search space by hierarchical clustering.** The architecture indices on leaves correspond to indices used in Pooling benchmark (Roshtkhari et al., 2023) (left) Tree learned from output vectors (our method) ; (middle) From vector encoding ; (right) From one-hot encoding.

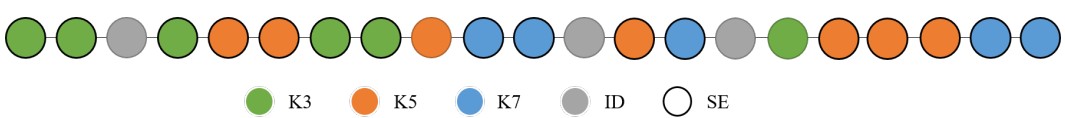

Figure 6: **Visualization of found architecture for ImageNet.** Different colors correspond to various kernel sizes and identity operation, while bold borders indicate SE module.

Table 7: **CIFAR10 results on NAS-Bench-Macro (Su et al., 2021a)** search space with several various rewards. Relative rewards are calculated according to Su et al. (2021a). The rewards can be can be calculated on either training or validation set.

| Search Structure | Metric | Reward Data | Reward Measure | Arch. | Best Acc. | Best Rank | Avg. Rank |
|---|---|---|---|---|---|---|---|
| Default Tree + Reg | rel. | train | loss | [22121222] | 92.74 | 85 | 97 |
| Learned Tree (ours) | rel. | train | loss | [22122220] | 92.78 | 61 | 67 |
| Default Tree + Reg | abs. | train | acc. | [22121210] | 92.55 | 227 | 278 |
| Learned Tree (ours) | abs. | train | acc. | [22110222] | 92.56 | 209 | 301 |
| Default Tree + Reg | rel. | val. | loss | [22222022] | 92.71 | 98 | 120 |
| Learned Tree (ours) | rel. | val. | loss | [21211220] | 92.76 | 71 | 95 |
| Default Tree + Reg | rel. | val. | acc. | [12222222] | 92.92 | 21 | 112 |
| Learned Tree (ours) | rel. | val. | acc. | [22212200] | 92.94 | 19 | 67 |
| Default Tree + Reg | abs. | val. | acc. | [22221200] | 92.86 | 34 | 54 |
| Learned Tree (ours) | abs. | val. | acc. | [22212220] | 93.13 | 1 | 6 |

## C.5 COMPLEXITY ANALYSIS

Considering $N$ architectures in the search space, the computational complexity to build the tree for our method is determined by two factors: the complexity of inference to obtain output vectors from the pre-trained supernet ($O(N)$) and the cost of distance calculation and clustering ($O(N^2)$). The cost of inference depends on the complexity of the architectures in the search space, while the output distance calculation depends on number of output classes. Therefore, the total complexity can be estimated as $aN^2 + bN$. We estimate that our method works best for $N < 10k$ and estimate values of $a = 1e - 8$ and $b = 0.01$ for our ImageNet experiments. Therefore, the cost of inference dominates the overall cost.

