# OpenReview forum: "Neural Architecture Search by Learning a Hierarchical Search Space"
_ICLR.cc/2025/Conference — Submitted to ICLR 2025_

### Official Review · Reviewer_c3Y4 · 2024-11-01

**Soundness:** 2
**Presentation:** 1
**Contribution:** 2
**Rating:** 3
**Confidence:** 3

**Summary:**

This paper presents a Neural Architecture Search (NAS) approach that leverages Monte Carlo Tree Search (MCTS) with a learned hierarchical search space. Instead of using a non-optimal, pre-defined hierarchical search order, this paper proposes to learn the branching by hierarchical clustering of architectures based on their similarity measured by the pairwise distance of output of architectures. The experiments on CIFAR10 and ImageNet demonstrate that the proposed approach yields better solutions than previous approaches.

**Strengths:**

This paper highlights the shortcomings of the previously used node independence assumption and demonstrates that too restrictive assumptions converge to worse solutions.

**Weaknesses:**

The weaknesses of this paper are the limited novelty, results not significant, and the unclear approach.

First, for the limited novelty and results not significant, this work improves previous works (Wang et al., 2021a; Zhao et al., 2021b) by replacing the model accuracy with the output vector. While the output vector provides more information for clustering architectures, the novelty is limited. Furthermore, the results only slightly outperform previous works (Su et al., 2021a; You et al. 2020; …), which is insignificant. In addition, making early tree nodes more discriminative is highly relevant to the partitioning or splitting problems in decision tree learning, which has been studied by many in the past (Costa and Pedreira, 2023).
Costa, V.G., Pedreira, C.E. Recent advances in decision trees: an updated survey. Artif Intell Rev 56, 4765–4800 (2023).

Second, the presentation has a lot of improvements, especially the approach. The proposed method is ambiguous and does not seem like MCTS. MCTS uses UCT to select child modes; however, the proposed method uses Boltzmann sampling with a UCB-like score as the parameter. The authors should justify whether this design follows the UCT foundations of balancing exploration and exploitation.
Most importantly, using $Acc(a_i)/n_i$ in Eq. 4 is weird. From the definition, the first term in the formula is the average reward (Eq. 1 in Kocsis & Szepesvári, 2006). However, Eq. 4 further divides the accuracy by the visit count. Since the accuracy $Acc(a_i)$ is already considered the average reward, it makes no sense. If this is not a typo, the authors should justify the correctness of such a design.

Please refer below for more questions.

**Questions:**

Questions related to the proposed MCTS procedure:
- As the tree is already constructed, does the algorithm still run selection from a single root node and then expand the known tree structure? Or does it just start sampling from the entirely constructed tree?
- The typical MCTS involves several phases (e.g., selection, expansion, simulation, backpropagation) per simulation, while it is unclear how the proposed procedures in Algorithm 1 are linked to these phases.
- It is mentioned that $C(a_i) = Acc(a_i)$ for architecture search in line 340. Does it mean that for supernet training, $C(a_i)$ is set to Eq. 4? It is unclear which parts use $Acc(a_i)$ or Eq. 4 in Algorithm 1.
- In Algorithm, $P_{train}$, $P_{search}$, and Eq. 5 are not defined.
- In line 358, it is mentioned that there is a warm-up period for uniform sampling, which is also not included in the typical MCTS routines (Kocsis & Szepesvári, 2006). As MCTS should already be able to balance exploration and exploitation, what is the purpose of adding such a warm-up period?
- In line 360, why is $C(a_i) < 1$ when nodes are visited?

Other comments related to typos and presentation issues:
- For Figure 1, it is difficult to understand why the subfigures "independent" and "joint" are drawn like this.
- For Figure 2, (b) and (c) use different styles to represent the tree structure, which should be normalized to the same.
- The section title "3.4 Sampling with conditional probabilities: Monte Carlo Tree Search" is confusing as this section does not seem to have any links to (the typical) MCTS.
- For Table 3, "the categorical vector representation" is not included in the discussion.
- Some terms are not consistently used, e.g., "Monte Carlo Tree Search" or "Monte-Carlo Tree Search"; "equation" or "eq." or "Eq.".
- Several typos, e.g., "Fig.3.4(a)", "and T the temperature term", "UTC".
- Placing Algorithm 1 in Appendix B lowers the readability. It would be more appropriate to include it in the main text, especially since the authors often refer to it with "see algorithm 1 in Appendix B".
- In Algorithm 1, the equations should be explicitly stated instead of mentioning "as in Eq. 1."

---

> ### Author Response · Authors · 2024-11-22
> **Response to Reviewer DxTU: Part 1**
>
> We thank the reviewer for their insightful review. We did our best to answer in a meaningful and honest way to all comments and misunderstandings. We believe that because of reviewer's comments, the new version of the manuscript is in a much better shape and hope the reviewer would give us a chance to read and discuss our answers and revise their score.
>
> In parts 1 and 2 we address the weaknesses of the paper and will answer the questions shortly.
>
> ### Novelty and Significance
>
> > this work improves previous works (Wang et al., 2021a; Zhao et al., 2021b) by replacing the model accuracy with the output vector. While the output vector provides more information for clustering architectures, the novelty is limited.
>
> We would like to clarify some points about the comparison of our work with Wang et al., 2021a and Zhao et al. 2021b. In the related work of the main paper, we did not explain explicitly the differences between our paper and those two “similar” works. The most relevant difference is that in our work, we aim at training a deep learning model (in particular a CNN), while jointly learning its best architecture. This is quite a challenging task, because it requires using an algorithm for finding promising architectures that should work online and with partially trained models.
>
> Therefore, the estimations of the architecture likelihoods are extremely noisy. Additionally, If the model samples the wrong configuration, this would influence the rest of the training and lead to worse results. In this setting, it is of paramount importance to quickly find promising architectures, as this will avoid sampling bad architectures which will negatively impact the training and potentially without any possibility of back-tracking.
>
> In contrast, the two papers mentioned above tackle the easier problem of finding the best architectures with a reduced set of samples, assuming that the recognition model is given, either with a supernet trained uniformly with all architectures (for Wang et al. 2021a), which we show does not perform well on our benchmarks, or by precomputed models as in BenchNAS-201 (for Zhao at al. 2021b).
> The only paper that does a joint training of the model and finds the most promising architecture with MCTS is Su et al., 2021a, and that’s the reason we compare directly and thoroughly with it, showing the advantages of our approach.
>
> > Furthermore, the results only slightly outperform previous works (Su et al., 2021a; You et al. 2020; …), which is insignificant.
>
> Our CIFAR and ImageNet tests show either better accuracy or lower computational cost or both compared to other methods. While the gains may seem small, the distribution of architecture performances for CIFAR10 benchmarks is narrow with many architectures performing well (e.g. for distribution of architectures in NAS-Bench-Macro see figure 7.c in [Su et al., 2021a]). In this sense, it is impossible to significantly improve the performance of the best final architecture numerically even if we find the best architecture benchmark. For example in table 4, while the best accuracy for our method is numerically close to MCTS + Reg, we are able to find the best architecture in the benchmark and our method consistently shows higher performance when considering the average accuracy and ranking.
>
> > In addition, making early tree nodes more discriminative is highly relevant to the partitioning or splitting problems in decision tree learning, which has been studied by many in the past
>
> We believe that also this point is due to the previous misunderstanding. For methods in which the estimation of the probabilities of each architecture is accurate, using a discriminative approach makes sense. Here, however, at the beginning of the training the estimation of the quality of each architecture is noisy, as the model has not been fully trained on that. Thus, directly using that to drive the sampling would lead to poor results (for instance the results of the uniformly sampled model). In contrast, a method that only considers the similarity of architecture outputs is not affected by the specific quality of the model. It considers only the semantic similarity among architectures on the output space and can lead to better performance. We are currently running experiments to show empirically that and will update the answer with results when available.

---

> ### Author Response · Authors · 2024-11-22
> **Response to Reviewer DxTU: Part 2**
>
> ### Presentation and Proposed Method
>
> > Second, the presentation has a lot of improvements, especially the approach. The proposed method is ambiguous and does not seem like MCTS. MCTS uses UCT to select child modes; however, the proposed method uses Boltzmann sampling with a UCB-like score as the parameter.
>
> The reviewer is correct that our method is not the typical MCTS as it skips the expansion stage. However, we followed the naming of previous works, such as [Su et al., 2021a, Wang et al., 2021a], that do not include the expansion stage due to the fact that the connection of nodes in the hierarchy is static and known prior. In addition, we introduce stochasticity with Boltzmann sampling, to allow a faster exploration of different configurations, to not bias the model to similar configurations multiple times.
>
> Thus, overall, we agree with the reviewer that there are quite some differences from the commonly used MCTS. However in our opinion the spirit of MCTS is maintained, and the name could help readers to situate the paper. We are open to changing the naming if reviewer think that this could improve the quality and readability of the paper.
>
> > Most importantly, using Acc(a_i)/n_i in Eq. 4 is weird....
>
> Thank you very much for pointing this out. We apologize for the typo. We corrected it and double checked the other equations.

---

> > ### Comment · Reviewer_c3Y4 · 2024-11-26
> >
> > I thank the authors for clarifying, which addresses some of my concerns.
> >
> > The clarification regarding novelty and significance is understandable, but the authors must heavily revise the paper to highlight the importance of this work. Other reviewers also raised similar concerns about novelty and significance, showing the necessity of improving this part.
> >
> > For MCTS, I agree with the authors that the spirit of MCTS is maintained. However, to avoid misleading, the authors should clearly present the modified procedure and use appropriate citations instead of Kocsis & Szepesvári (2006). Other reviewers also raise concerns about MCTS, e.g., Boltzmann sampling, so the authors should provide convincing evidence for using such a design.

---

> > > ### Author Response · Authors · 2024-11-28
> > > **Response to Reviewer DxTU: Part 3**
> > >
> > > ### Questions Regarding MCTS Procedure
> > >
> > > > As the tree is already constructed, does the algorithm still run selection from a single root node and then expand the known tree structure? Or does it just start sampling from the entirely constructed tree?
> > >
> > > It is true that the hierarchy (tree) structure is learned and fixed prior to the search. The algorithm still starts from the root node and samples child nodes from the probability distribution (eq. 2) using UCT (eq. 3). Then, in the backpropagation phase, it updates the probabilities associated with each node.
> > >
> > > > The typical MCTS involves several phases (e.g., selection, expansion, simulation, backpropagation) per simulation, while it is unclear how the proposed procedures in Algorithm 1 are linked to these phases.
> > >
> > > Similar to [Su et al., 2021a;Wang et al. 2021a], the expansion and simulation phase in the typical MCTS algorithm are skipped in our method. This is because the connectivity of the nodes in the search tree is static and in our opinion, skipping the expansion stage does not make a meaningful difference in the algorithm. In fact, the main difference in performing the expansion vs. using a fully expanded tree is that the algorithm will start the update of all visited nodes from the beginning without waiting to expand a node. For the simulation, as we use Boltzmann sampling and initially the choice between branches are initialized uniformly, in the beginning the exploration of a new branch would be the same as in the simulation. However, we update the entire path as all the branches are already expanded.
> > >
> > > > It is mentioned that $C(a_i) = Acc(a_i)$  for architecture search in line 340. Does it mean that for supernet training, $C(a_i)$ is set to Eq. 4? It is unclear which parts use $Acc(a_i)$ or Eq. 4 in Algorithm 1.
> > >
> > > We reworked the algorithm for more clarity using directly the equations where possible. To better distinguish between training and architecture search,, we edited the sentence with $\lambda=0$ for the search phase (output of algorithm). There was a mis-alignment between Eqs and the algorithm. Now it should be correct, with everything properly defined.
> > >
> > > > In line 358, it is mentioned that there is a warm-up period for uniform sampling, which is also not included in the typical MCTS routines (Kocsis & Szepesvári, 2006). As MCTS should already be able to balance exploration and exploitation, what is the purpose of adding such a warm-up period?
> > >
> > > Also this point stems from the main difference with classical MCTS problems in which the estimation of the reward for a given configuration is fixed and available from iteration 0. Here, instead the learning of our recognition model (a CNN) is happening in parallel with the exploration of the tree. Thus, a warm-up period is needed to avoid early biases due to a too early estimation of the quality of the models.
> > >
> > > > In line 360, why is $C(a_i) < 1$ when nodes are visited?
> > >
> > > Because $C(a_i)$ represents the accuracy of the architecture on the validation mini-batch, and thus it will always be equal or smaller than one.

---

> > > > ### Author Response · Authors · 2024-11-28
> > > > **Response to Reviewer DxTU: Part 4**
> > > >
> > > > ### Other Questions
> > > >
> > > > > For Figure 1, it is difficult to understand why the subfigures "independent" and "joint" are drawn like this.
> > > >
> > > > As the variables are binary, for the sake of clarity we considered the joint distribution as a 3D table. Thus, for the independent, we marginalize the joint into one dimension. We would appreciate any other suggestion the reviewer has to improve this figure.
> > > >
> > > > > For Figure 2, (b) and (c) use different styles to represent the tree structure
> > > >
> > > > The goal for Fig. 2 (b) was to highlight branching on the operation, while for (c) we wanted to show the clustering. We have been thinking quite a bit about the best way to show this, and Fig. 2 it is the best representation we could propose. We are open to any other suggestion for improvement and would appreciate it greatly. We also changed the letters for the architectures to a_i, to be in line with the rest of the paper.
> > > >
> > > > > The section title "3.4 Sampling with conditional probabilities: Monte Carlo Tree Search" is confusing as this section does not seem to have any links to (the typical) MCTS.
> > > >
> > > > The reviewer is right. We mainly discuss tree design here and not MCTS, so we changed the title to “Sampling with conditional probabilities: Tree Search”
> > > >
> > > > > For Table 3, "the categorical vector representation" is not included in the discussion.
> > > >
> > > > We added that to the discussion. The “categorical vector representation” is the equivalent categorical encoding of one-hot encoding obtained from the adjacency matrix. Since we deal with feature map resolutions in this case, the use of categorical vectors is reasonable.
> > > >
> > > > > Some terms are not consistently used... Several typos...
> > > >
> > > > We apologize, we reviewed the text and corrected the inconsistencies and typos.
> > > >
> > > > > Placing Algorithm 1 in Appendix B lowers the readability.
> > > >
> > > > We agree that it would be best to include it in the main text, and the reason we included it to the appendix was the lack of space in the main paper. Unfortunately, we could not find any part of the paper that could easily go to appendix without harming the presentation of our work and results.
> > > >
> > > > > In Algorithm 1, the equations should be explicitly stated instead of mentioning "as in Eq. 1."
> > > >
> > > > We edited the algorithm by using the explicit form of equations (where possible) as the reviewer suggested, which has improved the reliability of the algorithm.

---

### Official Review · Reviewer_5Vyd · 2024-11-01

**Soundness:** 2
**Presentation:** 2
**Contribution:** 2
**Rating:** 6
**Confidence:** 2

**Summary:**

This paper challenges the commonly assumed node independence in Neural Architecture Search (NAS), which may limit both efficiency and performance. To address this, the authors propose a Monte Carlo Tree Search (MCTS) method incorporating a learned hierarchical tree structure, built with agglomerative clustering based on model output distances, to improve NAS effectiveness. Experiments are conducted on NAS benchmarks for CIFAR-10 and ImageNet image classification tasks.

**Strengths:**

* The paper introduces an approach by addressing node dependencies to improve NAS efficiency.
* Leveraging the UCT (Upper Confidence bounds applied to Trees) approach, the authors further utilize a learned tree structure to reduce the reliance on manually crafted search space designs.
* The paper provides ablation studies to analyze the effects of the proposed method in more depth.

**Weaknesses:**

* The abstract may benefit from significant revision. Currently, it primarily highlights MCTS fundamentals and suggests general applicability, but the paper is focused on a NAS-specific task that utilizes MCTS and related techniques to enhance NAS performance. The abstract and introduction appear inconsistent in conveying the core contribution and scope.
* The rationale behind using model output distances to construct the tree structure and improve NAS is not clearly discussed, and the method itself lacks detail. This part should be the core of the paper, yet there is minimal explanation in the main text.
* While resource constraints may be a factor, it remains unclear whether the method scales well for large networks, which are particularly relevant in NAS applications. The experiments mainly validate that the learned tree provides slight improvements but do not assess scalability in larger search spaces.

Minor Comments
* Line 306: check around "Fig.3.4(a)"

**Questions:**

* How does this method compare with other state-of-the-art NAS techniques, such as those in the Neural Architecture Transfer (NAT) series?
* What insights or theoretical basis underlie the decision to use model output distances for improving NAS performance? (I already assume this will be addressed in a revision in my score.)

---

> ### Author Response · Authors · 2024-11-28
> **Response to Reviewer 5Vyd**
>
> We thank the reviewer and appreciate their insightful comments and suggestions. Below, we address the concerns and questions raised:
>
> ### Weaknesses
>
> > The abstract may benefit from significant revision. Currently, it primarily highlights MCTS fundamentals and suggests general applicability, but the paper is focused on a NAS-specific task that utilizes MCTS and related techniques to enhance NAS performance. The abstract and introduction appear inconsistent in conveying the core contribution and scope.
>
> We refined the abstract to clarify the goal and the scope of the paper and make it more consistent with the rest of the paper. While the paper focuses on the specific application of NAS, the learning meaningful ordering of data is general and beneficial in many domains where training data is sequential but the order of which the data is sampled vary.
>
> > The rationale behind using model output distances to construct the tree structure and improve NAS is not clearly discussed, and the method itself lacks detail. This part should be the core of the paper, yet there is minimal explanation in the main text.
>
> While the ultimate objective of NAS is to find the model with the highest accuracy, when naively using the supernet, the accuracy of architectures can have low correlation with ground truth accuracy. This makes it an unreliable metric for tree construction. The output vector on the other hand is a more informative summary metric for an architecture than the accuracy, as it defines a semantic relationship between architectures that is independent from the actual accuracy of the model. The key idea is to build a hierarchy in which distances between architectures have semantic meaning in the class space. This is also reflected in the fact that using encodings (table 3) shows better results than using uniform sampling. We added more details to section 4.(Tree Design) about tree design.
>
> > While resource constraints may be a factor, it remains unclear whether the method scales well for large networks, which are particularly relevant in NAS applications. The experiments mainly validate that the learned tree provides slight improvements but do not assess scalability in larger search spaces.
>
> It is true that using our method in the current form works best in smaller search spaces. However, there are possible ways to deal with larger search spaces, such as using FLOPs as a training-free proxy that we used for ImageNet (similar to [Su et al., 2021a]) or using other zero-cost proxies [Abdelfattah et al, 2021] to prune the search space. Our results on ImageNet shows that our method can still be feasible in larger search spaces with these techniques. To add more clarity to the paper, we included this limitation to the conclusion.
>
> > minor comment: Line 306: check around "Fig.3.4(a)"
>
> Corrected.
>
> ### Questions
>
> > How does this method compare with other state-of-the-art NAS techniques, such as those in the Neural Architecture Transfer (NAT) series?
>
> In general, although both NAS and NAT aim to automate the design of neural architecture, they have different approaches and goals. NAS focuses on discovering novel architectures for a task from scratch, while NAT focuses on adapting a pre-trained architecture to a new task and leveraging the knowledge for more efficiency. This makes direct comparison of them difficult.
>
> The motivation of the work was not to outperform the state-of-the-art, rather to provide a new understanding of NAS sampling approaches and highlight the advantages of hierarchical search spaces for NAS and propose solutions to challenges in their design, showing good performance in a limited amount of time/computation.
> In this sense, we have included the methods that are highly relevant to ours in our comparisons.
>
> > What insights or theoretical basis underlie the decision to use model output distances for improving NAS performance? (I already assume this will be addressed in a revision in my score.)
>
> Addressed above in weaknesses.
>
> ### references
>  [Abdelfattah et al, 2021]: Zero-Cost Proxies for Lightweight NAS, ICLR 2022.

---

> > ### Comment · Reviewer_5Vyd · 2024-12-02
> >
> > After reviewing the revised version, I found the explanation of the method and its underlying ideas to be clearer compared to the previous submission. I have no further questions and am willing to increase the score to 6. However, due to my limited knowledge in the NAS domain, I will maintain my confidence level at 2.

---

### Official Review · Reviewer_xUxx · 2024-11-04

**Soundness:** 3
**Presentation:** 3
**Contribution:** 2
**Rating:** 6
**Confidence:** 4

**Summary:**

This paper proposes a method for supernet sampling for neural architecture search using Monte-Carlo Tree Search (MCTS). After an initial phase of supernet training, the method uses similarity distances between architecture outputs and hierarchical clustering to build a search tree, then continue the supernet training by sampling from this tree using MCTS.

**Strengths:**

-The paper is overall well written.
-The methodology is well-explained and the contributions are clearly defined, the paper is well-placed in the literature.
-While not theoretically justified, the idea of learning the Monte-Carlo tree is promising.
-The experimental results are convincing on the ImageNet dataset.

**Weaknesses:**

-The method still requires initial supernet training using uniform sampling before being able to build the tree, which is known to be computationally heavy.
-The overall contribution seems incremental, as it is mainly a new way to construct a Monte-Carlo tree for supernet sampling.
-For the experiment on the pooling dataset, the authors explain that this extremely small search space of 36 architectures is challenging because the initial supernet training shares weights between architectures with different pooling configurations. Given that the proposed method discriminates architectures by comparing the outputs after supernet pre-training, I wonder how the method is able to find a more efficient representation of the tree if the weights themselves are not optimal. Furthermore, the classical sampling methods (uniform, Boltzmann…) are unable to find the best architecture out of 36? How many samples are performed? The results, while in line with the results of [1], seem surprising and the paper could benefit from a more thorough explanation.
-There are several typos and the writing is overall unclear in Section 5.1.
-Is the Boltzmann sampling over UCT in Section 4.2 necessary? The UCT formula already offers a trade-off between exploration and exploitation. If it is necessary, then an ablation study could be useful.
-The following claim : “Different from other works such as Wang et al. (2021a) and Zhao et al. (2021b) that use the model accuracy directly for the tree design, the output vector provides more information for clustering architectures” seems unsupported.
-Building the search tree requires building a hierarchical clustering. As the authors use the pairwise distance matrix of all architectures in the search space over a mini-batch, the complexity of building this hierarchical clustering is O(n^2) complexity. For large search spaces, this could be very inefficient. A comparative complexity analysis of the proposed method would be welcome.

[1] : Javan et al., Balanced Mixture of SuperNets for Learning the CNN Pooling Architecture, 2023

**Questions:**

The paper proposes an interesting idea, is mainly well-written and shows some good results on benchmark datasets. As written in the weaknesses section, there are several avenues for clarification and improvements on the paper.

---

> ### Author Response · Authors · 2024-11-28
> **Response to Reviewer xUxx: Part 1/3**
>
> We thank the reviewer for their valuable comments and suggestions. We are thrilled that the reviewer found our work well written and promising. We address the concerns and questions below:
>
> ### Weaknesses
>
> > The method still requires initial supernet training using uniform sampling before being able to build the tree, which is known to be computationally heavy
>
> While it is true that our method requires pre-training the supernet to construct the tree, full training is not required. In Figure 3 we show that training until convergence is not required to get the best results and training for 1/2  of iterations outperforms MCTS+reg and 2/3 of iterations already achieves the same result as full training (Figure 2 (left)). Furthermore, even with the pre-training we obtain a lower search time than several other methods (e.g. Table 5 for ImageNet) .
>
> We tested several potential ways to build the tree without supernet pre-training in our “ablation-alternative branching” by using untrained supernet, and various encodings to calculate distances for clustering. While they don’t perform as well as using supernet outputs, using weighted encodings still outperform uniform, Boltzmann and vanilla MCTS (comparing table 3 with table 1). Finding better ways to reduce the computational cost of building the tree, while obtaining a high performance will be explored in future work.
>
> > The overall contribution seems incremental, as it is mainly a new way to construct a Monte-Carlo tree for supernet sampling
>
> We believe that our contribution is not incremental because:
>
> We are the first to learn the tree structure for Monte-Carlo sampling during the joint training of the sampling probabilities (to focus the training on the best configuration) and the recognition model (a deep convolutional neural network). Previous methods were learning the tree structure (Wang et al. 2021a, Zhao at al. 2021b) while assuming the recognition model given (either a supernet pre-trained uniformly or models precomputed in BenchNAS-201 ). The only method that learns both recognition model and tree probabilities is Su et al., 2021a, which does not learn the tree structure.
>
> Learning the structure of the tree in such a joint learning is quite challenging because, in the beginning of the training all estimations are extremely noisy and using directly the model accuracy to separate the search space would not work. Instead, using a hierarchical clustering on the output representation just measures the similarity of different architectures and works well even in such challenging settings. We added a paragraph in section 5.2 (Tree partitioning with accuracy) that shows that accuracy is not a good metric for building the tree structure when the recognition model is poor.
>
> These two contributions are quite new, and can foster research in this challenging but very useful setting. We are aware that we did not explain those points clearly enough in the original version paper, but we believe they are important and should be considered by reviewers.
>
> > For the experiment on the pooling dataset...  Given that the proposed method discriminates architectures by comparing the outputs after supernet pre-training, I wonder how the method is able to find a more efficient representation of the tree if the weights themselves are not optimal.
>
> In general, weight sharing (used in popular one-shot NAS methods such as DARTS and its  variants), while very efficient, can have a negative effect on the outcome of the NAS, due to interference among architectures. In this particular benchmark, the weight sharing is full (all architectures always share 100% of weights). To tackle this, [Javan et al., 2023] uses multiple supernets to reduce the weight sharing from the start; instead, our method also reduces the weight sharing, but gradually with a single model. By being able to focus on high performing architectures, the weights will be effectively only shared by smaller and smaller numbers of architectures.
>
> Even if the supernet has poor correlation with the optimal architecture due to the full weight sharing, the supernet is still used only to cluster similar architectures and not to rank those architectures. This is the very reason why methods based directly on the accuracy of the supernet would not work as well as ours. To show this empirically, we ran a basic experiment to recursively partition search space based on accuracy. More specifically to design the binary tree by halving the search space into “good” and “bad” regions and performing the search. The results we obtained (best 90.85, avg. 90.49) is significantly diminished compared to our method.

---

> > ### Author Response · Authors · 2024-11-28
> > **Response to Reviewer xUxx: Part 2/3**
> >
> > > Furthermore, the classical sampling methods (uniform, Boltzmann…) are unable to find the best architecture out of 36?  How many samples are performed? The results, while in line with the results of [1], seem surprising and the paper could benefit from a more thorough explanation.
> >
> > We experimented with uniform sampling and Boltzmann sampling to optimize the number of iterations and temperature term (both fixed and linear temperature) and tested these methods for various number of iterations (up to 1200 epochs) on this benchmark.
> >
> > For uniform sampling, because of the full weight sharing, the model minimizes a loss that gives the same importance to each architecture. Thus, the final model would be the model that performs best on all architectures, while we look for the model that performs the best on the best architecture. Thus, there is a mismatch between the loss and our objective.
> >
> > For Boltzmann sampling, the training  learns the importance of each architecture, and samples accordingly. Thus, in theory this sampling should be able to find a good architecture. However, as with this sampling we consider the joint distribution of the architectures, for each sample we update only the probability of one architecture at the time. Thus, updates on sampling probabilities are very slow and cannot catch-up with the model updates. This mismatch can cause the training to gets stuck in a spurious local minimum. That motivated our approach, in which the hierarchical estimation of the architecture’s probabilities is faster and can keep up with the model learning and avoids spurious local minimums.
> >
> > > There are several typos and the writing is overall unclear in Section 5.1
> >
> > We apologize for the typos, we edited section 5.1 to improve clarity.
> >
> > > Is the Boltzmann sampling over UCT in Section 4.2 necessary? The UCT formula already offers a trade-off between exploration and exploitation. If it is necessary, then an ablation study could be useful.
> >
> > Boltzmann sampling (eq. 2) is used to relax UCT and introduces stochasticity to the method. Similar to Su et al., 2021a, the rationale was that incorporating Boltzmann sampling allows sampling more diverse architectures by providing soft probabilistic exploration. This is quite helpful because in contrast to a classic MCTS, here we learn jointly the recognition model (the CNN) and the tree, therefore the exploration of the tree is used for training our model. Thus, more variety in the exploration helps to train a better model. We added more detail in sec. 4.(search and training).
> >
> > > The following claim : “Different from other works such as Wang et al. (2021a) and Zhao et al. (2021b) that use the model accuracy directly for the tree design, the output vector provides more information for clustering architectures” seems unsupported.
> >
> > Unfortunately, the sentence we used seems to be misleading; in the sense that implies the only difference with our method is using accuracy vs. the output vector. We would like to clarify that our method has more fundamental differences with approaches proposed by Wang et al. (2021a) and Zhao et al. (2021b). The most relevant difference is that we aim at training a deep learning model (CNN) while jointly estimating the architectures, which is challenging as it should work online with partially trained models.
> >
> > Therefore, the estimations of the architecture likelihoods are extremely noisy. Additionally, If the model samples the wrong architecture, this would influence the rest of the training and lead to worse results. In this setting, it is of paramount importance to quickly find promising architectures, as this will avoid sampling bad architectures which will negatively impact the training and potentially without any possibility of back-tracking.
> >
> > In contrast, the two papers mentioned above tackle the easier problem of finding the best architectures with a reduced set of samples, assuming that the modeling is given, either with a supernet trained uniformly with all architectures (for Wang et al. 2021a), which we show does not perform well on our benchmarks, or by precomputed models as in BenchNAS-201 (for Zhao at al. 2021b).
> >
> > The only paper that does a joint training of the model and finds the most promising architecture with MCTS is Su et al., 2021a, and that’s the reason we compare directly and thoroughly with it, showing the advantages of our approach.

---

> > > ### Author Response · Authors · 2024-11-28
> > > **Response to Reviewer xUxx: Part 3/3**
> > >
> > > >  Building the search tree requires building a hierarchical clustering. As the authors use the pairwise distance matrix of all architectures in the search space over a mini-batch, the complexity of building this hierarchical clustering is O(n^2) complexity. For large search spaces, this could be very inefficient. A comparative complexity analysis of the proposed method would be welcome.
> > >
> > > It is true that using our method in the current form works best in smaller search spaces. However, there are possible ways to deal with larger search spaces, such as using FLOPs as a training-free proxy that we used for ImageNet (similar to (Su et al., 2021a)) or using other zero-cost proxies (Abdelfattah et al, 2021) to prune the search space. Our results on ImageNet shows that our method can still be feasible in larger search spaces with these techniques. To add more clarity to the paper, we included this limitation to the conclusion.
> > >
> > > In the supplementary material, we added a section about complexity analysis. There we show that, while it is true that the computation of the pairwise matrix is quadratic, the main computational cost is the inference of the recognition model. For larger search spaces, this quadratic cost can be an issue more in terms of memory storage than computation.
> > >
> > > ### references
> > > Abdelfattah et al, 2021: Zero-Cost Proxies for Lightweight NAS, ICLR 2022.

---

### Author Response · Authors · 2024-11-28
**New version of manuscript**

We thank the reviewers for their detailed reading of the manuscript and valuable comments. We have responded to each reviewer individually and uploaded a new version of the paper that addresses the reviewers’ concerns.

The additions and changes are shown in blue in the new version. The summary of changes to the manuscript are the following:


- Added a sentence to the abstract to connect it more to the actual problem we are solving, as suggested by Rev. 5Vyd
- Reshaped the last part of the introduction to explain clearly the difference between our approach and two MCTS related papers and the use of clustering on the network class output
- Changed related works on the same two papers, to better explain the differences
- Updated fig. 2 to have the same letter for the architectures, as suggested by Rev. c3Y4
- Updated Tree design in our method, better explaining the advantages of our clustering for the tree partitioning as asked by Rev. 5Vyd
- Improved Search and Training in our method, to acknowledge that our MCTS does not use all the phases of a standard MCTS, and uses Boltzmann sampling to improve the exploration. Added also more context about what are the actual phases of the training. Rev. c3Y4 asked to move the algorithm in the main paper, but unfortunately we did not have space.
- Updated Eq. 3 and 4, to fix a typo and to be more in line with the algorithm in the supplementary material
- Improved the presentation of the pooling search space in section 5.1 as asked by Rev. xUxx
Added an ablation to compare our clustering with a partitioning based on accuracy as asked by Rev. xUxx
-  Added more explanations for tab. 3 in the text
- Added limitations in conclusion
- Improved algorithm in supplementary material as asked by Rev. c3Y4
- Added complexity analysis to the supplementary as asked by Rev. xUxx

---

### Author Response · Authors · 2024-12-01
**Summary of Rebuttal**

We would like to thank the reviewers for their valuable comments which greatly helped us in improving our paper. We have responded to each reviewer individually and uploaded a new version of the paper that addresses the reviewers’ concerns. We would like to summarize our answers to the main concerns raised by reviewers:

***

**Experimental Results:**  The experiments in our paper were carried out on two CIFAR10 NAS benchmarks and ImageNet mobilenet-like search space. The distribution of the accuracy of architectures in those benchmarks is narrow, with many architectures performing well (for example for distribution of architectures in NAS-Bench-Macro see figure 7.c in Su et al., 2021a). Therefore, numerical gains in terms of accuracy may seem small. However, our method is able to find architectures close to the best (on CIFAR10) or the best (on NAS-Bench-Macro in table 4) and significantly reduce the search cost on ImageNet compared to similar methods.

**Contribution:** We understand reviewers' concerns about the proposed contribution. However, some of those issues were due to a not accurate presentation of the related work. In particular, reviewers considered (Wang et al. 2021a) and (Zhao at al. 2021b) comparable to our work, but with a different strategy to partition the search space with a tree. However, those two papers do not learn the recognition model while finding the optimal architecture as we do. Instead they use MCTS on a given and fixed recognition model (Uniformly trained CNN for (Wang et al. 2021a) and BenchNas-201 for (Zhao at al. 2021b)).

This makes the problem we our tackle quite different than those two and that require different solution and not really comparable. For instance, the choice of using a clustering approach for the tree (in contrast to using directly the model accuracy as in (Su et al., 2021a; Wang et al. 2021a)) is due to the fact that the initial model has poor performance and accuracy is a poor proxy for the real model performance. Thus, partitioning with that would lead to suboptimal results. We added some explanations about that in the presentation of the method and an additional experiment to confirm this intuition. Instead, considering proximity in an unsupervised manner, our clustering algorithm takes into account the model similarities and differences without considering accuracy and leads to better results.
The only paper performing the joint learning of the recognition model and search of the optimal architecture as us is (Su et al., 2021a), and we analyze and compare explicitly and thoroughly with them.

**Limitations:** Our model in its current form works best for search space of roughly <10k, due to the quadratic complexity of the similarity matrix. We have analyzed this in Appendix C.5, showing that for small search spaces the main computational cost is the computation of the recognition model output, which is linear in the number of architectures. For larger search space, the complexity of the similarity matrix is an issue, not much in terms of computation, but more in terms of memory storage.

In general, there are possible ways to deal with larger search spaces. For instance, on ImageNet tests, we used FLOPs as a simple training-free proxy for performance (similar to Su et al., 2021a) to prune the search space. Other techniques such as zero-cost proxies (Abdelfattah et al., 2022) can be combined with our method to eliminate unpromising architectures.

**references**

Su et al., 2021a: Prioritized Architecture Sampling with Monte-Carlo Tree Search, CVPR 2021.

Wang et al. 2021a: Sample-efficient neural architecture search by learning actions for monte carlo tree search, 2021.

Zhao at al. 2021b: Multi-objective optimization by learning space partitions, ICLR 2022.

Abdelfattah et al., 2022: Zero-Cost Proxies for Lightweight NAS, ICLR 2022.

***

We hope our answers satisfy reviewers' questions, doubts and comments and look forward to further discussions and clarification. We also hope the reviewers take into account the improved manuscript and responses during the rebuttal and reconsider their scores.

---

### Author Response · Authors · 2024-12-02
**Request for feedback and revaluation of the scores!**

Dear reviewers and AC,
we answered all your comments and clarification, updated the paper and provided a summary of the most critical points.
We believe the new version of the paper is substantially improved and we would like your feedback and comments on that before the end of the discussion period, to give us a chance to raise your scores!
Thank you!

---

### Meta-Review · Area_Chair_jy1v · 2024-12-22

**Metareview:**

In this work a MTCS approach to neural architecture search. The method uses similarity distances between the architecture outputs and a hierarchical clustering algorithm to build a search tree. The experimental evaluation is conducted on CIFAR and ImageNet classification tasks.

Some reviewer’s concerns included the expensive training of the supernet procedure, and the writing of the abstract that focuses on MCTS which shows discrepancy with the general theme of the paper that is centered around NAS. Finally, the main issue raised by the reviewers was the limited novelty and unconvincing experimental evaluation showing sufficient gains for the proposed approach. This is not mitigated by a clear explanation of the approach that justifies the novel algorithmic choices and may mitigate concerns about the performance gap.

**Additional Comments On Reviewer Discussion:**

After the reviewer discussion round was done, agreement was not reached as to whether this work constitutes a sufficient contribution for publication at ICLR. Issues raised range from the unconvincing experimental evaluation showing little to no gains w.r.t. existing approaches to the lack of novelty of the approach.

---

### Decision · Program_Chairs · 2025-01-22

Reject